# High-resolution Nanopore methylome-maps reveal random hyper-methylation at CpG-poor regions as driver of chemoresistance in leukemias

Alberto Magi [1,2,8 ✉], Gianluca Mattei[1,8], Alessandra Mingrino[3], Chiara Caprioli[4,5], Chiara Ronchini[4], Gianmaria Frigè[4], Roberto Semeraro[3], Davide Bolognini[3], Alessandro Rambaldi [5,6], Anna Candoni [7], Emanuela Colombo [4], Luca Mazzarella [4] & Pier Giuseppe Pelicci [4,5 ✉]

Aberrant DNA methylation at CpG dinucleotides is a cancer hallmark that is associated with the emergence of resistance to anti cancer treatment, though molecular mechanisms and biological significance remain elusive. Genome scale methylation maps by currently used methods are based on chemical modification of DNA and are best suited for analyses of methylation at CpG rich regions (CpG islands). We report the first high coverage whole-genome map in cancer using the long read nanopore technology, which allows simultaneous DNA-sequence and -methylation analyses on native DNA. We analyzed clonal epigenomic/genomic evolution in Acute Myeloid Leukemias (AMLs) at diagnosis and relapse, after chemotherapy. Long read sequencing coupled to a novel computational method allowed definition of differential methylation at unprecedented resolution, and showed that the relapse methylome is characterized by hypermethylation at both CpG islands and sparse CpGs regions. Most differentially methylated genes, however, were not differentially expressed nor enriched for chemoresistance genes. A small fraction of under-expressed and hyper-methylated genes at sparse CpGs, in the gene body, was significantly enriched in transcription factors (TFs). Remarkably, these few TFs supported large gene-regulatory networks including 50% of all differentially expressed genes in the relapsed AMLs and highly-enriched in chemoresistance genes. Notably, hypermethylated regions at sparse CpGs were poorly conserved in the relapsed AMLs, under-represented at their genomic positions and showed higher methylation entropy, as compared to CpG islands. Analyses of available datasets confirmed TF binding to their target genes and conservation of the same gene-regulatory networks in large patient cohorts. Relapsed AMLs carried few patient specific structural variants and DNA mutations, apparently not involved in drug resistance. Thus, drug resistance in AMLs can be mainly ascribed to the selection of random epigenetic alterations at sparse CpGs of a few transcription factors, which then induce reprogramming of the relapsing phenotype, independently of clonal genomic evolution.

[1] Department of Information Engineering, University of Florence, Florence, Italy. [2] Institute for Biomedical Technologies, National Research Council, Segrate, Milano, Italy. [3] Department of Experimental and Clinical Medicine, University of Florence, Florence, Italy. [4] Department of Experimental Oncology, IEO European Institute of Oncology IRCCS, Milano, Italy. [5] Department of Oncology and Hemato-Oncology, University of Milan, Milan, Italy. [6] Azienda Socio-Sanitaria Territoriale Papa Giovanni XXIII, Bergamo, Italy. [7] Clinica Ematologica, Azienda Sanitaria Universitaria Integrata di Udine, Udine, Italy. [8] These authors contributed equally: Alberto Magi, Gianluca Mattei. ✉email: albertomagi@gmail.com; piergiuseppe.pelicci@ieo.it

DNA modification of 5-methylcytosine (5mC) at CpG dinucleotides is the most frequent epigenetic change in cancers. They include genome-wide hypo-methylation in regions of low CpG density, in conjunction with site-specific hypermethylation of CpG-rich regions (CpG islands). It is generally thought that DNA hyper-methylation promotes tumorigenesis by silencing tumor suppressor genes or genes controlling genomic instability, cell adhesion or apoptosis, while genome-wide DNA hypomethylation causes chromosomal instability, de-repression of retrotransposons, and occasionally oncogene overexpression[1]. Aberrant patterns of methylation and associated gene-expression signatures have also been documented in relapsed and therapy-resistant samples in a variety of cancer-types, yet their function in the establishment of drug-resistance remains unclear[2–5]. Currently available methods of genome-scale methylation analyses are based on bisulfite conversion of cytosine (C) to thymine (T) followed by next- generation sequencing (whole-genome bisulfite sequencing—WGBS; reduced representation bisulfite sequencing—RRBS)[6]. They have, however, many limitations that restrict resolution of the methylome, including DNA degradation, due to bisulphite treatment, PCR-induced CpG artifacts and inability to distinguish 5mC from 5-hydroxymethylcytosine, the first product in the demethylation of 5mC[6]. Furthermore, C-to-T conversion creates divergence with the reference genome, thus reducing alignment, sequencing coverage and mapping of low complexity CpG regions. Notably, ~90% of the human genome is characterized by regions of low density CpGs (1–3 CpG/100 bp), with most of the remaining showing densities of ≥5 CpG/100 bp, including CpG islands[6]. The last few years have seen the emergence of third-generation sequencing technologies, based on nanopore sequencing[7], which allow analyses of single nucleic-acid molecules and produce sequences in the order of tens to hundreds kilobases (kb). The basic principle is the transit of single DNA filaments through a nanoscopic pore with concomitant measurement of their effects on the electric current of a connected electrode[8,9]. Current variation can be then used to infer sequence-base content and to recognize base modifications, such as 5mC[10], thus allowing quantitative assessment of methylation in native DNA, with the potential of bypassing main limitations of bisulphite sequencing. Furthermore, the ultra-long length of nanopore reads facilitate de novo assembly and calling of structural DNA variants (SV), permitting concomitant analyses of genomic and epigenomic alterations. However, after the initial report of the human genome and methylome by nanopore sequencing in 2018 and 2017, respectively[10,11], the potential of long-read sequencing in cancer genomics and epigenomics has been only preliminarily explored. Long-reads analyses have clearly shown higher accuracy and sensitivity to detect SVs in the DNA or plasma of a few breast-cancer[12] and lung-carcinoma samples[13]. As well, informative DNA methylation profiles have been obtained in breast cancer cell lines[10] and series of hepatocellular carcinoma[14] and brain-tumor[15] samples.

Here we present the first study that exploits high-coverage nanopore whole-genome sequencing (WGS) to analyze simultaneously genomic and epigenomic alterations in cancer, using sample-pairs at diagnosis and relapse of Acute Myeloid Leukemias (AMLs). AML is a high-mortality disease with a 5-year survival of <30%[16]. Despite incorporation of targeted drugs and immunotherapies in recent years, standard of care remains chemotherapy, and relapse of drug-resistant disease is the main cause of mortality[16]. Molecular mechanisms underlying the acquisition of the drug-resistant phenotype, however, are not clear[17].

We developed a novel computational approach properly devised to identify DNA-methylation alterations in sample pairs (PoreMeht), which allowed definition of differential methylation at unprecedented resolution ( > 99% of CpGs) and analyses of both CpG islands and sparse CpGs. We report that the drug-resistance phenotype in AMLs is supported by the selection of random epigenetic alterations at sparse CpGs of few transcription factors that induce transcriptional reprogramming of the relapsing phenotype.

## Results

**Nanopore reads coupled to a novel computational method extends analyses of differential methylation to sparse CpGs (outside CpG islands).** We analyzed tumor samples at diagnosis (T) and relapse (R) from three AML patients (UD5, UD10 and AML2) who received standard chemotherapy and relapsed with chemoresistant disease (supplementary Table 1). DNA from each sample was sequenced on the GridION instrument using R9.4 flow cells (Supplemental Material), obtaining 5–10 million reads per sample (supplementary Fig. 1a) for a total of ~60 − 100 billion bps after quality filtering (supplementary Fig. 1c), average read size of 10–15 Kb (supplementary Fig. 1b) and sequencing coverage of 20–30x (supplementary Fig. 1d).

Analyses of 5mC, using Nanopolish[10], predicted methylation of ~22.6 millions of CpG groups, for a total ~28 millions of CpG sites (84% CpG groups contained one CpG site; 10% two CpG sites) (supplementary Table 2) that represent ~99% of all CpGs of the human genome[18]. Remarkably, between 75 and 95% of all CpG groups were supported by 10 reads (Fig. 1a), a percentage much higher than obtained by classical bisulfite-conversion methods (50-70% for WGBS and 20–25% for RRBS)[19].

Percentage of methylation ($\beta$) was calculated at each CpG group as the ratio between methylated and analyzed CpGs (supplementary Fig. 2). For all the six samples, $\beta$ had an average value of 0.8 (supplementary Fig. 3) and showed the typical bimodal distribution of methylation data[20,21], with the two modes located at 0 for hypo-methylated and 1 for hyper-methylated CpGs (Fig. 1b). $\beta$ values were highly correlated between the six samples, especially for matched sample pairs (T-R, supplementary Fig. 4l).

Differential methylation between R and T samples was calculated by computing $\beta$ values differences at each CpG group ($\Delta\beta = \beta_R - \beta_T$). $\Delta\beta$ takes values in the range [−1,1], where $\Delta\beta > 0$ or < 0 indicates, respectively, hyper- or hypo-methylation of the relapse- vs. diagnosis-samples. For all three AML pairs, $\Delta\beta$ values showed a trimodal distribution with hyper- ($\Delta\beta > 0.3$) or hypo- ($\Delta\beta < −0.3$) methylation of only ~3% of all CpGs (Fig. 1c).

Since genomic distribution of DNA methylation follows strong local-patterns, differential methylation across genomic regions (differentially methylated regions, DMRs) are considered statistically and biologically more informative than differentially-methylated CpGs[22]. In WGBS datasets, DMRs are identified as groups of consecutive CpGs with concordant hypo- or hyper-methylation, upon segmentation of $\Delta\beta$ values of spatially ordered cytosines[23,24]. This approach, however, is suboptimal for the analyses of long-read methylation data, which are instead characterized by large low-density CpG regions alternated with small high density CpG regions.

We thus developed a novel tool (PoreMeth) based on a heterogeneous form of the shifting level model (SLM)[25], which integrates distances between consecutive CpGs in the segmentation algorithm. PoreMeth is fed with $\Delta\beta$ values and generates genomic segments with increased (hyper-) or decreased (hypo-) methylation levels between two samples (Fig. 1d), which are then evaluated for statistical significance using the Wilcoxon-rank sum test (Supplemental Material). PoreMeth was validated in silico by analyses of a reported synthetic dataset of differential methylation[23] (Supplementary Figs. 5–11) and shown to

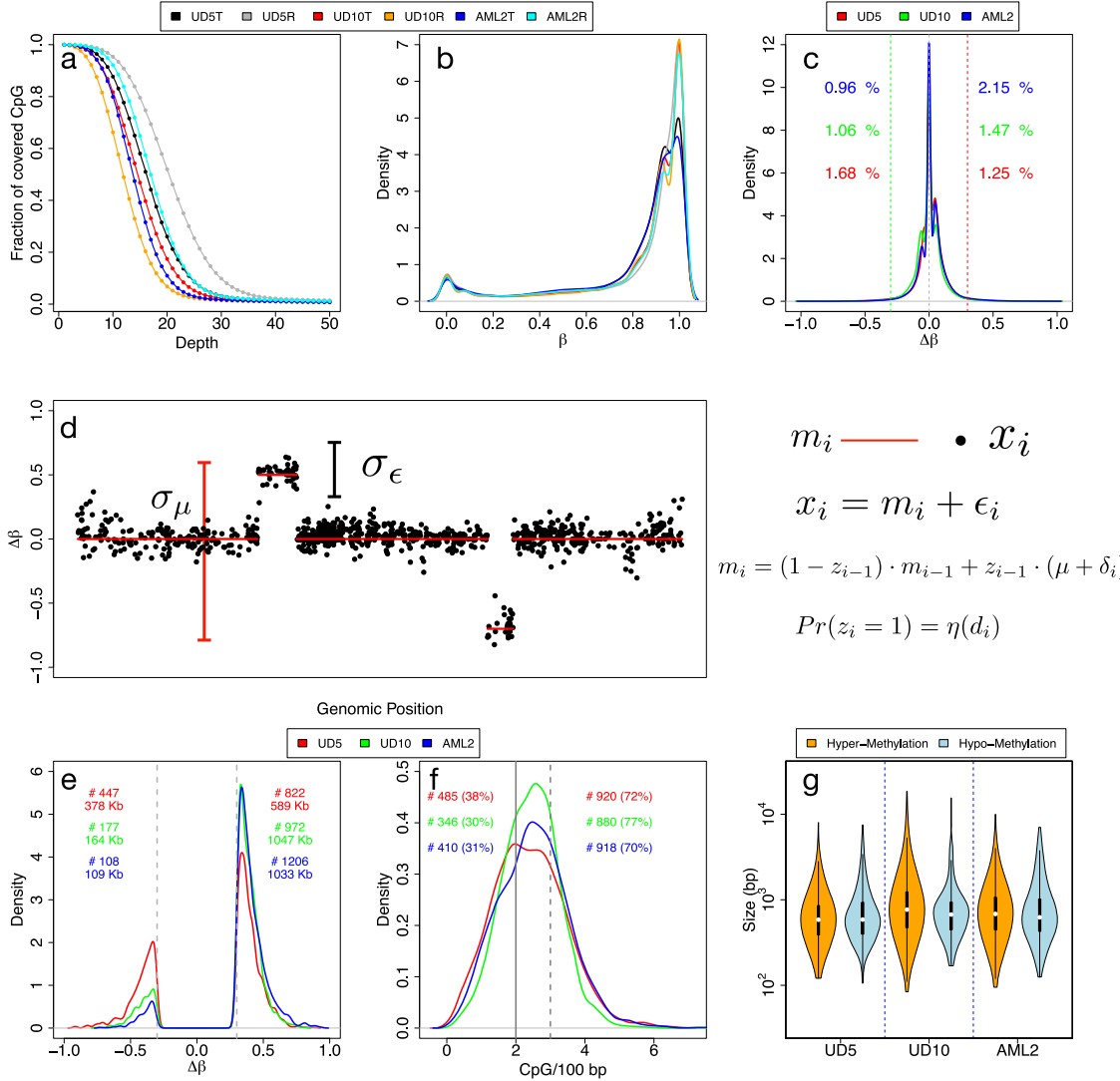

**Fig. 1 Methylation frequency distribution and Poremeth. a** Shows the fraction of covered CpG groups as a function of sequencing depth. **b** Reports the density distribution of $\beta$ values, while **c** the density distribution of $\Delta\beta$ values. **d** Shows the SLM model. $\Delta\beta$ values are modeled as the sum of two independent stochastic processes $m_i$ and $\epsilon_i$, where $m_i$ is the unobserved mean level and $\epsilon_i$ is normally distributed white noise ($\epsilon_i \sim N(0, \sigma_\epsilon^2)$). $m_i$ changes according to a geometric distribution (with parameter $\eta$) incrementing its value by the normal random variable $\delta_i$ ($\delta_i \sim N(0, \sigma_\mu^2)$). To take into consideration the sparse nature of CpGs along the genome, the probability ($\eta$) to jump from two different state $m_i$ ($Pr(z_i)=1$) depends the distance $d_i$ between consecutive CpGs. Distribution of the $\Delta\beta$ values of the DMRs detected by Poremeth on the three pairs of AML samples (**e**, red for UD5, green for UD10 and blue for AML2). Text on left (right) side of the plot reports the total number (#) and total size (in kb) of hypomethylated (hypermethylated) DMRs. **f** Shows CpG density distribution of the DMRs detected by Poremeth on the three pairs of AML samples (red for UD5, green for UD10 and blue for AML2). Vertical continuous line indicates the resolution limit of WGBS ($\leq 2$ CpG/100 bp), while vertical dotted line indicates ERRBS resolution limit ($\leq 3$ CpG/100 bp). Text on left (right) side of the the plot reports the total number (#) and percentage (%) of DMRs detected by PoreMeth with CpG density $\leq 2$ CpG/100 bp ($\leq 3$ CpG/100 bp). The violin plots in (**g**) show the size distribution of hyper-methylated (orange) and hypo-methylated (light-blue) DMRs detected by PoreMeth in the analysis of the three AML samples.

outperform currently used methods (Metilene and BSmooth)[23,24] in terms of both precision and recall, especially for highly noisy data (supplementary Fig. 12).

We then applied PoreMeth to the three AML sample pairs. DMRs were assigned to segmented regions using a $\Delta\beta$ cutoff of 0.3 (>0.3, hyper-methylated; < 0.3, hypo-methylated) and a significance of $p < 0.05$ (supplementaryData 1). We identified 1,269 DMRs for UD5 (947 kb of genomic regions), 1,149 for UD10 (1,211 kb) and 1,314 for AML2 (1,142 kb), including both hyper- and hypo-methylated DMRs (Fig. 1e).

All three relapsed samples showed a significantly larger fraction of hyper-methylated vs. hypo-methylated DMRs, in terms of numbers (822 vs 447 for UD5, 972 vs 177 for UD10 and 1206 vs

108 for AML2) and total size (589 kb vs 378 kb for UD5, 1047 kb vs 164 kb for UD10 and 1033 kb vs 109 kb for AML2) (Fig. 1e). Due to the high-resolution of the Nanopore maps, ~30% of DMRs showed a CpG density $\leq 2$ CpG/100 bp (e.g. the resolution limit of WGBS) and ~ 70%$\leq 3$ CpG/100 bp (resolution limit of ERRBS) (Fig. 1f), suggesting that the identified DMRs involve both high density CpG regions (CpG islands or CGIs) and sparse CpGs (NoCGI)[26].

Average size of DMRs was 700–800 bp for all three AML pairs, with a distribution ranging from hundreds bp to tens kb and no significant differences between hyper- and hypo-methylated (Fig. 1g). Thus, long-read sequencing coupled to our novel computational method allows definition of differential

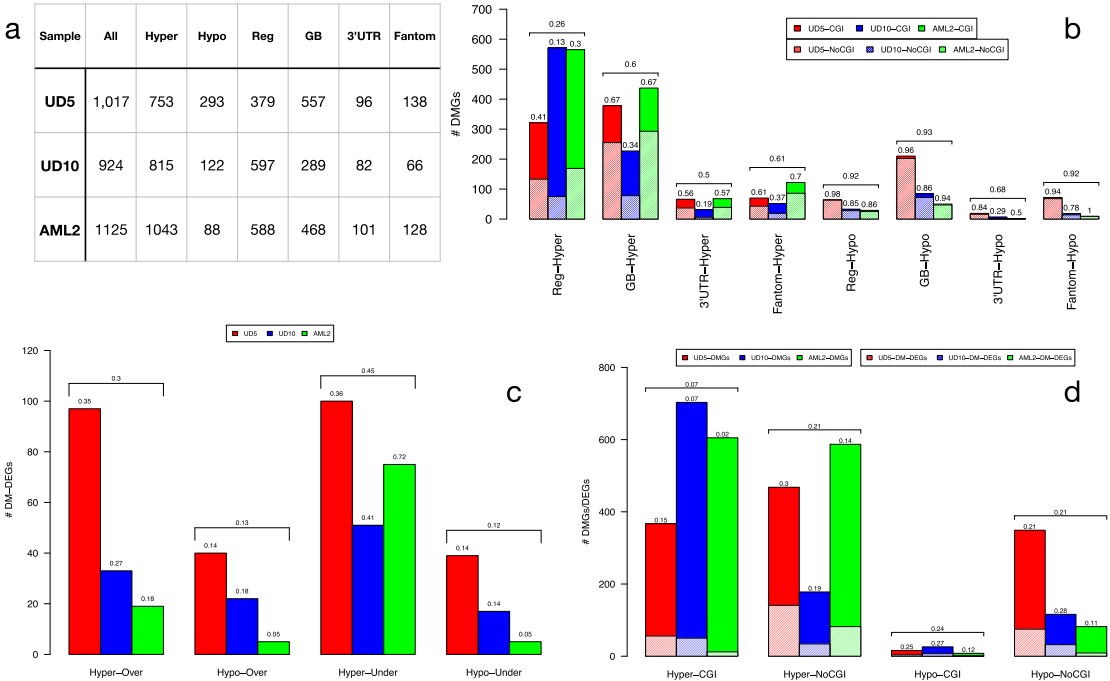

**Fig. 2 DMRs functional classification.** Table in (**a**) reports the number of genes affected by all, hyper-methylated (Hyper) and hypo-methylated (hypo) DMRs. For each patient we also report number of genes with DMRs overlapping 5' regulatory region (Reg, including 1–5 kb from TSS, promoters, first exons and 5'UTRs), gene-body (GB, including internal introns and exons), 3'UTR and enhancers (genes predicted by FANTOM to be regulated by enhancers). The discrepancy between total number of genes (All) and the sum of Reg, GB and 3'UTR is due to the fact that few genes have DMRs in more than one genic element. The barplot of (**b**) reports the total number of DM genes with DMRs at different genomic elements (5' regulatory region, Reg, internal introns and exons, GB, 3'UTR and enhancers of Fantom Genes). with DMRs in all regions (All, both inside and outside CGIs), inside CpG islands (CGI) and outside CpG islands (NoCGI) at different genomic elements (5' regulatory region, Reg, internal introns and exons, GB, 3'UTR and enhancers of Fantom Genes). Textured bars show the number of DMGs with DMRs in sparse CpG regions (NoCGI). Numbers above bars show the percentage of DMGs in sparse CpG regions. Horizontal brackets above each group of three bars summarize average percentages of the three samples. The barplot of (**c**) reports the total number of DM-DEGs for the four regulatory categories: over-expressed genes with hyper-methylated DMRs (Hyper-Over), over-expressed genes with hypo-methylated DMRs (Hypo-Over), under-expressed genes with hyper-methylated DMRs (Hyper-Under), under-expressed genes with hypo-methylated DMRs (Hypo-Under). Numbers above each bar show the relative percentage of DM-DEGs with respect to all DM-DEGs for each sample. Horizontal brackets above each group of three bars summarize average percentages within the three samples. The barplot of (**d**) report the number of hyper- and hypo-methylated DM-DEGs with respect to total number of DMGs in CGIs and sparse CpG regions (NoCGI). Textured bars show the number of DM-DEGs. Numbers above each bar show the percentage of DM-DEGs with respect to DMGs. Horizontal brackets above each group of three bars summarize average percentages within the three samples.

methylation at unprecedented resolution, extending analyses of CpG islands, as achieved until now, to sparse CpGs, which represent about 40% of all DMRs in the tested sample-pairs.

**Differentially methylated genes are hypermethylated at CpG-islands or sparse CpGs and only occasionally differentially expressed.** To evaluate the functional impact of differential methylation, we annotated each DMR to genes and genic element by using the Refseq gene models for 5' regulatory regions (Reg), gene bodies (GB), and 3'UTR and to datasets of regulatory elements such as CpG islands, DNase I hypersensitive sites (DHSs), transcription-factor binding sites (TFBS) and the FANTOM5 database for enhancers (see methods).

As expected from CpG density distribution, 26, 65 and 45% DMRs (for UD5, UD10 and AML2 respectively) overlapped CGIs, while the remaining were located outside, at sparse CpGs (Supplementary Table 3).

DMRs at CGIs were almost exclusively hyper-methylated (96-98% in the three samples) while DMRs at sparse CpGs were hypo- or hyper-methylated (54–46, 63–37 and 86–14% in the three samples) and represented the only regions of hypo-methylation in the relapsed samples. Most DMRs mapped within annotated genes (72, 72 and 77% for UD5, UD10 and AML2,

respectively) and overlapped DHSs (95, 97 and 96%) and TFBS (83, 89 and 88%), with a small fraction also overlapping enhancers (16, 10 and 16%) (Supplementary Table 3).

Numbers of genes involved by DMRs (differentially-methylated genes; DMGs) were ~1000 per patient (Fig. 2a). 30–50% showed DMRs at 5' regulatory regions (Reg), 30–50% at gene-bodies (GB) and ~10% at 3'untranslated regions (3'UTR). DMRs at 5' regulatory regions, 3'UTR and enhancers formed sharp peaks, while DMRs at internal exons and/or introns were uniformly distributed across the entire gene-body (Supplementary Fig. 13). Approximately 100 additional genes were predicted by FANTOM5 as regulated by hyper- or hypo-methylated enhancers (138, 66 and 128 for UD5, UD10 and AML2, respectively).

Most DMGs were hyper-methylated (~70, ~85 and ~93% in UD5, UD10 and AML2, respectively, Fig. 2a). In the genes hyper-methylated at 5' regulatory regions, DMRs mainly involved CpG islands (~74% across the three samples: 59, 87 and 70% for UD5, UD10 and AML2). Involvement of sparse CpGs was instead predominant in the genes hyper-methylated at gene-bodies or enhancers (~60% across the three samples). Hypo-methylated genes were almost entirely associated with DMRs overlapping sparse CpGs (~90%), regardless of their position within genes

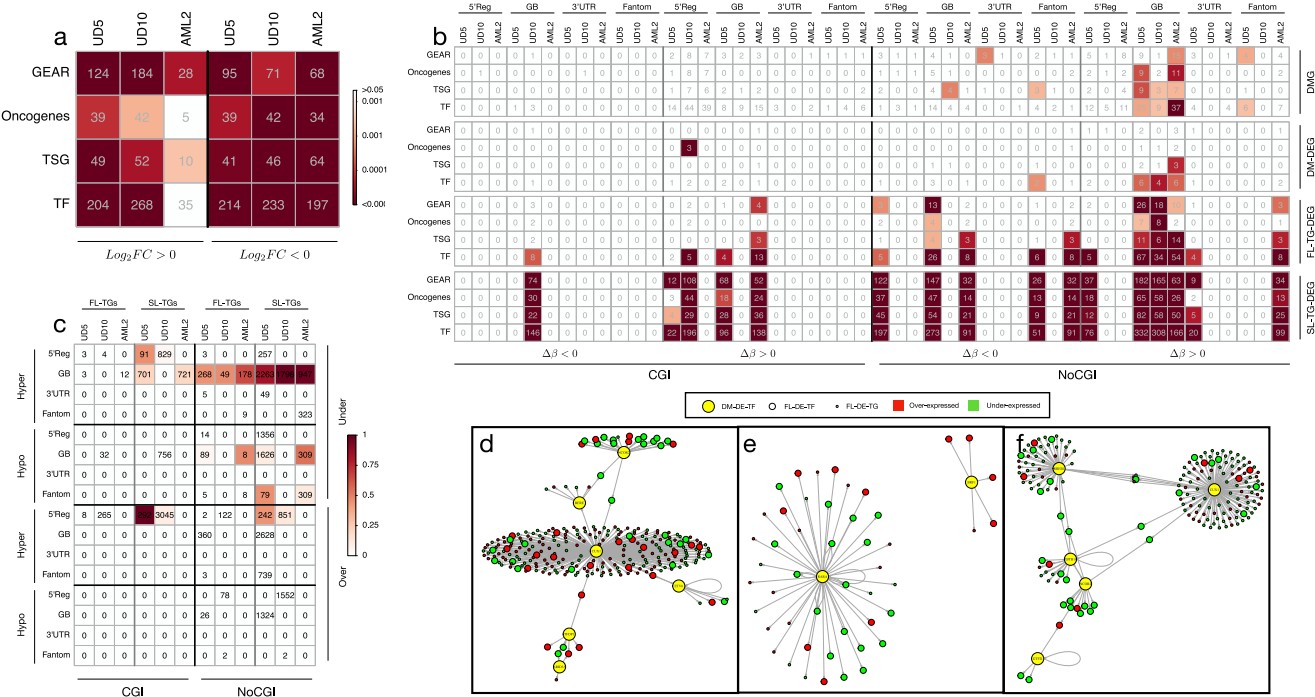

**Fig. 3 Impact of DMRs on gene expression. a, b** Show the results of ORA on DEGs (**a**), DMGs, under-expressed DM-DEGs, FL- and SL-TG-DEGs (**b**) for GEAR, Oncogenes, TSG and TF categories. Each cell of the plots reports the number of genes for each category. For both panels a-b color of each cell reflects the statistical significance expressed by Fisher exact test as in the color legend. **c** Reports the total number of DE-TGs of the first- (FL-) and second-level (SL-) regulatory cascade induced by over- and under-expressed TFs with hyper- or hypo-methylated DMRs (DM-DE-TF) in CpG islands (CGI) and sparse CpG regions (NoCGI) at different genomic elements (5′ regulatory region, Reg, internal introns and exons, GB, 3′UTR and enhancers of Fantom Genes). In each cell of the table is reported the number of DE-TGs for each regulatory category. For each sample, color of each cell reflects the proportion of DE-TGs shared with the other two samples. **d–f** Show GRNs induced by under-expressed DM-DE-TFs with DMRs at sparse CpG regions of the gene-body for UD5 (**d**), UD10 (**e**) and AML2 (**f**). Yellow large-size nodes are DM-DE-TFs, medium-size nodes are FL-DE-TFs and small-size nodes are FL-DE-TGs. Green nodes represent under-expressed TGs, while red nodes over-expressed TGs.

(Fig. 2b). Thus, the relapse methylome is prevailing hyper-methylated at CpG islands of 5′-regulatory regions or at sparse CpGs of gene-bodies.

To investigate the impact of DMRs on gene expression we performed triplicate RNA-sequencing analyses (RNAseq) of the six AML samples, and analysed differential gene-expression between relapse vs diagnosis samples using the DESeq2 tool[27]. We identified 3997, 4677 and 1759 differentially expressed genes (DEGs; adjusted $p$-value < 0.05 and absolute $log_2FC > 0.5$, Supplementary Data 2) in UD10, UD5 and AML2, respectively, with different ratios of over- and under-expressed genes (2044 vs 1953 in UD5; 2,890 vs 1,787 in UD10 and 495 vs 1,264 in AML2) (Supplementary Table 4).

Detectable RNAseq signals were found in ~70% of DMGs (75, 61 and 66% in UD5, UD10 and AML2, respectively) with ~ 15% also showing differential expression in sample-pairs (DM-DE genes: 23, 12 and 8% in UD5, UD10 and AML2, respectively) (Supplementary Fig. 14 and Supplementary Table 5). DM-DE genes showed either classical (hypo-methylation/over-expression or hyper-methylation/under- expression) or alternative (hyper-methylation/over-expression or hypo-methylation/under- expression) epigenetic correlations, with hyper-methylated and under-expressed genes the most frequently occurring (~45%), followed by hypermethylated and over-expressed genes (~30%) (Fig. 2c).

Analyses of the association between differential DMG expression and $\Delta\beta$ values at DMRs showed correlation between increased methylation and decreased expression of DMGs with DMRs at 5′ regulatory regions, much weaker at 3′UTRs and absent at gene-bodies, as reported in ref. 28, irrespective of the presence of DMRs at CGIs or sparse CpGs (Supplementary

Figs. 15–16). Remarkably, hyper-methylated genes with DMRs in sparse CpGs showed a higher fraction of DEGs with respect to those with DMRs in CGIs (21 vs 7%) (Fig. 2d), especially for DMRs at 5′-regulatory and gene-body regions (Supplementary Figs. 17–18). Thus, only 10–20% of the DM genes at relapse are also differentially expressed. They are most frequently hyper-methylated and under-expressed and accounts for only 2–5% of all DEGs (Supplementary Table 5).

**Under-expressed genes with hyper-methylation at sparse CpGs are enriched in transcription factors.** To investigate the contribution of differential methylation to chemoresistance, we first characterized relapse-specific transcriptional patterns. Over- and under-expressed DEGs were largely overlapping across the three samples (40–70% of DEGs shared among 2–3 samples) (Supplementary Fig. 19), suggesting that the relapse phenotype is characterized by common modules of transcriptional reprogramming.

To functionally characterize these modules, we performed over-representation analyses (ORA) against a collection of cancer-related pathways (selected from KEGG[29]), tumor suppressor genes (TSGs) and oncogenes (selected by COSMIC[30]), transcription factors (TFs)[31] and genes associated with drug resistance (GEAR, a database that contains genetic associations with 148 anti-cancer drugs across 952 cancer cell lines[32]). Each relapse AML sample showed over-representation of over- and under-expressed DEGs in TFs, drug-resistance genes, cancer-mutated genes (oncogenes and TSGs) (Fig. 3a) and several cancer-related pathways (mTOR, MAPK, ErbB, FoxO, TNF and HIF-1 signaling, apoptosis) (Supplementary Fig. 20).

In order to test the functional effect of DMRs at each genomic feature, we then performed ORA of DM and DM-DE genes, distinguishing those with differential methylation overlapping CGIs or sparse CpGs at different genic elements. DM and DM-DE genes with DMRs overlapping CpG islands showed no significant over-representation of any of the tested datasets, regardless of methylation status or expression (Fig. 3b and Supplementary Figs. 21 and 23). The same was observed for the hypo-methylated DM (Supplementary Fig. 21) and DM-DE (Supplementary Fig. 22) genes at sparse CpGs. Hyper-methylated DM-genes at sparse CpGs, instead, were enriched in TSGs and TFs (Fig. 3b) and a few cancer-pathways in all three patients (Supplementary Fig. 22). Under-expressed, but not over-expressed, DM-DE genes at sparse CpGs were enriched in TFs (Fig. 3b and Supplementary Fig. 22). Most notably, enrichments were restricted to DM and DM-DE genes with hypermethylated DMRs overlapping sparse CpGs at gene bodies (Fig. 3b).

Thus, the three relapsed AMLs share common transcriptional patterns, characterized by deregulated expression of oncogenes/TSGs, TFs and chemoresistance genes. DM-DE genes, instead, showed significant enrichment of under-expressed TFs with hyper-methylated DMRs overlapping sparse CpGs at gene-bodies. The effect of DMRs on the expression of these DM-DE-TFs is reported in Supplementary Figs. 23–35. Chemo-resistance genes were not significantly over-represented, suggesting that differential methylation plays a marginal role to the establishment of the chemoresistance phenotype.

**A few transcription factors under-expressed and hyper-methylated at sparse CpGs induce large gene-regulatory networks that are highly-enriched in chemoresistance-associated genes.** We hypothesized that the selective pressure of drug treatments is exerted on the transcriptional targets (TGs) of the hyper-methylated and under-expressed TFs (DM-DE-TFs). To identify putative TGs of DM-DE-TFs we interrogated RegNetwork, a database containing experimentally-observed or predicted transcriptional or post-transcriptional regulatory relationships[33], and crossed the output with our dataset of relapse-specific DEGs (Supplementary Fig. 37).

The DM-DE-TFs of the three samples ($n = 26$, 12 or 8 in UD5, UD10 or AML2) identified, respectively, a total of 689, 501 and 187 DE-TGs, which may represent the first-level of the gene-regulatory network (GRN) perturbed by DMRs (Supplementary Fig. 38). Strikingly, for all three patients, the largest GRN (268, 49 and 178 DE-TGs in the three samples) was supported by under-expressed DM-DE-TFs hyper-methylated at sparse CpGs of gene-bodies (Fig. 3c). The other categories showed none or smaller GRNs, usually present in one or two patients and supported by DMRs involving sparse CpGs (Fig. 3c and Supplementary Figs. 39–40).

Remarkably, the few under-expressed DM-DE-TFs hyper-methylated at sparse CpGs of gene-bodies showed direct TF-DNA interactions with >50% of the DE-TGs of first-level GRNs, strongly supporting a direct effect of these TFs in deregulating the inferred GRNs (see Methods and Supplementary Table 6).

Inspection of first level GRNs showed that in all three AMLs a large proportion of TGs (mainly those regulated by under-expressed DM-DE-TFs hyper-methylated at sparse CpGs of gene-bodies) were again TFs (first-level differentially expressed TFs—FL-DE-TFs: 24%, 59% and 29% in UD5, UD10 and AML2, Supplementary Fig. 40). Computation of all potential TGs of FL-DE-TFs revealed the existence of second-level GRNs (SL-DE-TGs), which were significantly larger than first-level GRNs, with the largest again supported by under-expressed DM-DE-TFs hyper-methylated at sparse CpGs in gene-bodies (Fig. 3c–f and

Supplementary Figs. 41–42). Globally, 26, 12 and 8 DM-DE-TFs potentially affect the transcription activity of 69% (3210, UD5), 88% (3512, UD10) and 55% (969, AML2) of all DEGs (Supplementary Fig. 38).

In all first- and second-level GRNs, the percentages of over- or under-expressed DE-TGs ranges between 30 and 70% (Supplementary Figs. 43–46), suggesting that the regulatory cascade induced by DM-DE-TFs has both activation and repression activity. Strikingly, 6, 2 and 5 TFs (in UD5, UD10 and AML2) hyper-methylated at sparse CpGs of gene-bodies potentially contributed to the deregulation of ~50% (Supplementary Fig. 44) of all DEGs in the three samples ($n = 2263$, 1798 and 947). Of them, a large fraction was in common across 2 or 3 samples (45%, 52% and 52%) (Fig. 3c and Supplementary Figs. 47–48).

Thus, a few hyper-methylated and under-expressed TFs with DMRs at sparse CpGs of gene-bodies potentially modify transcription of thousands of genes in each of three relapsed AMLs (Fig. 3d–f), suggesting that transcriptional deregulation induced by these DMRs propagates and amplifies through subsequent levels of GRNs converging to the disruption of similar or even identical pathways in the different samples.

To evaluate the impact of GRNs on the chemoresistance phenotype, DE-TGs were subjected to ORA using the same datasets as above. Results showed that among first-level GRNs, DE-TGs induced by DM-DE-TFs hyper-methylated at sparse CpGs of gene-bodies were over-represented in TSGs, TFs and drug-resistance genes (Fig. 3b) and some cancer-related pathways (Wnt, MAPK, HIF-1, Cell cycle and apoptosis; Supplementary Fig. 49) in all the three patients. Pathway over-representation became significantly more prominent in second-level GRNs for all the analyzed datasets (Fig. 3b and Supplementary Fig. 51). Notably, drug-resistance genes, which were not significantly enriched in DM or DM-DE genes, were instead highly over-represented in both first- and second-level GRNs (Fig. 3b and Supplementary Figs. 49 and 51).

Together, these data suggest that transcriptional reprogramming of the relapsing phenotype, including chemoresistance, can be largely ascribed to regulatory perturbations supported by a few TFs hyper-methylated at sparse CpGs of their gene-bodies.

Most notably, analyses of DEGs found in other cohorts of treated AML patients[34,35] (see Methods), showed, for all three patients, a significant overlap with first- and, second-level GRNs and all DEGs (Supplementary Fig. 55), suggesting the existence of common modules of transcriptional reprogramming in chemo-resistant AMLs.

**Hyper-methylated DMRs at sparse CpGs are poorly conserved across samples, are under-represented at their genomic positions and show high methylation-entropy.** We then investigated mechanisms of DMR accumulation in the relapsed samples. Analyses of the genomic distribution of DMGs across the three AML-samples showed a large extent of overlap among 2 or three samples for the DM-genes with DMRs located at CpG islands (39% 30% and 24% for hyper-methylated DMGs in UD5, UD10 and AML2 respectively, Fig. 4a). The overlap was most prominent for hyper-methylated DMRs at 5'-regulatory regions or gene bodies, either over- or under-expressed (Supplementary Figs. 44–45). Surprisingly, DM-genes with DMRs located at sparse CpG showed no or marginal overlap across the three samples, suggesting different mechanisms of DMR generation at CpG islands or sparse CpGs (Fig. 4a and Supplementary Figs. 53–54).

We then computed the observed vs expected ratio for the occurrence of DMRs at any given genomic region. Hyper-methylated DMRs overlapping CpG islands showed significant

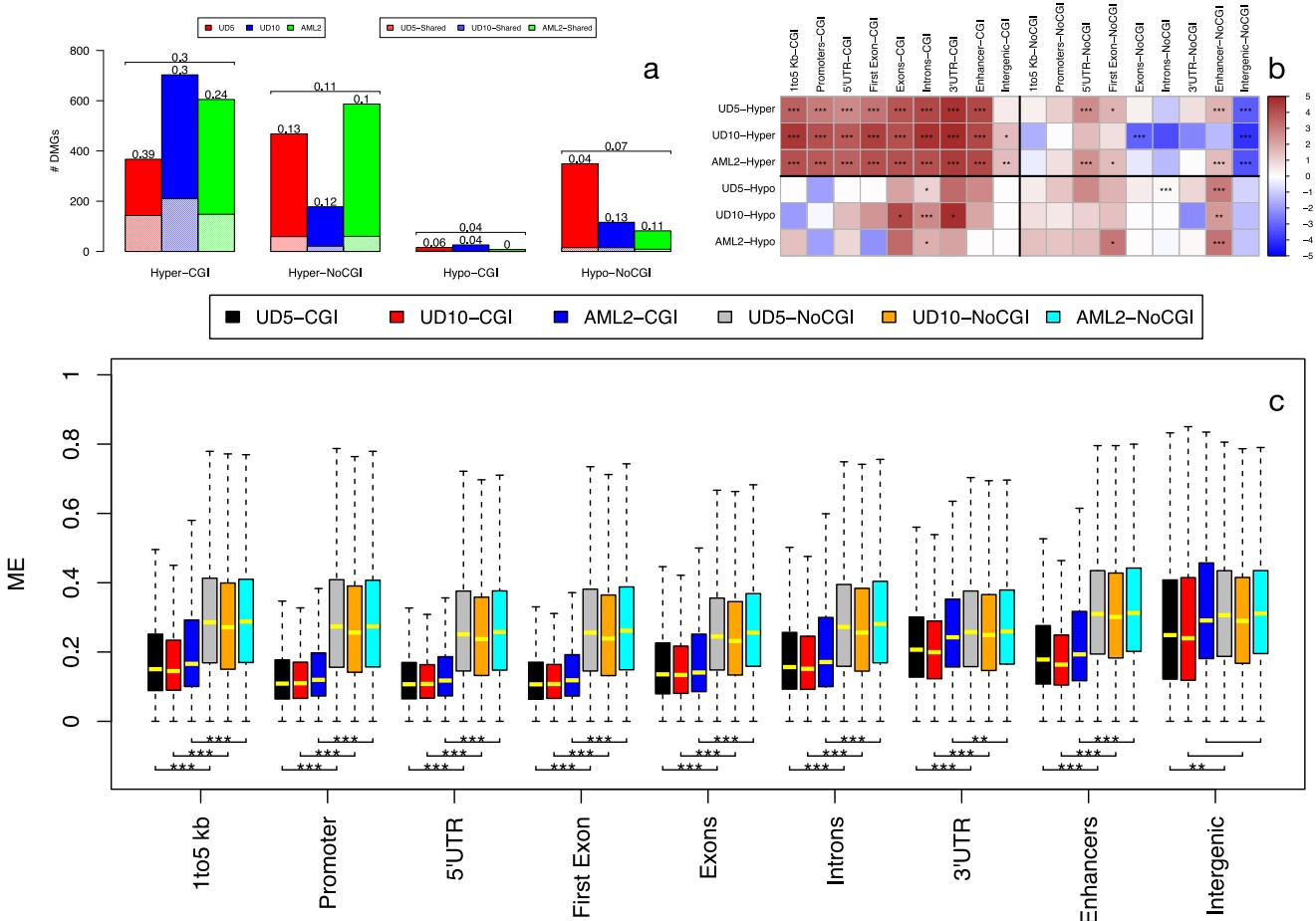

**Fig. 4 Mechanisms of DMRs selection.** For each AML sample, the barplot of (**a**) reports the fraction of DMGs shared with other samples. Results are reported for hyper- and hypo methylated DM genes with DMRs in CGIs and in sparse CpG regions (NoCGI). Numbers above bars show the percentage of shared DMGs. Horizontal brackets above each group of three bars summarize average percentages of the three samples. Observed vs expected analysis (**b**) is reported for hyper- (orange) and hypo-methylated (light-blue) DMRs overlapping genic elements (1–5 kb from TSS, Promoter, 5'UTR, First Exon, internal exons and introns and 3'UTR) inside CpG islands (CGI) and outside CpG islands (NoCGI). Over- and under-representation is defined by red and blu intensities according to color legend. Asterisks over each cell indicate statistical significance (*$p$-value < 0.01, **$p$-value < 0.001, ***$p$-value < 0.0001) of observed vs expected ratio. **c** Show the distribution of methylation entropy in different genomic features inside and outside CGIs for the three sample at diagnosis. Asterisks indicate statistical significance (calculated with $t$-test, *$p$-value < 0.01, **$p$-value < $10^{-5}$, ***$p$-value < $10^{-10}$) of the difference between ME in CGIs and outside CGIs. for each genomic feature.

over-representation at all genic and regulatory regions (5'UTRs, promoters, first exons, internal exons, introns and enhancers). Hyper- and hypo-methylated DMRs at sparse CpGs, instead, did not show significant over- or under- representation (with the exception of enhancers), suggesting a markedly higher level of stochasticity for the latter (Fig. 4b).

To confirm this hypothesis, we measured the randomness of DNA methylation at CpG sites (methylation entropy) for each genic and regulatory element, using the Shannon entropy[36]. DNA-methylation entropy is minimal when all cells share same DNA-methylation patterns, maximal when instead all possible patterns are equally represented. Strikingly, DNA-methylation entropy of all genic and regulatory elements was markedly higher at sparse CpGs, as compared to CpG islands (Fig. 4c), confirming high DNA-methylation stochasticity at CpG-poor regions.

Thus, formation of hyper-methylated DMRs at CpG islands or sparse CpGs is the consequence of highly ordered or stochastic processes, respectively, suggesting the involvement of active, enzymatically-driven vs random processes of DNA methylation. Chemotherapy may then select randomly-generated methylation patterns at sparse CpGs associated with a resistance phenotype.

**Long-reads sequencing identifies few, sub-clonal and patient-specific structural-variants in relapsed AMLs.** Finally, we investigated whether chemotherapy also selects genomic variants associated with the resistance phenotype, by using nanopore (for SVs) and whole-exome sequencing (WES) data (for single nucleotide variants—SNVs, and small insertions/deletions—InDels).

To identify relapse-specific SVs with nanopore datasets we used NanomonSV on the three AML pairs and, after visual inspection of alignment signatures, we identified 5, 3 and 4 SVs in UD5, UD10 and AML2 samples, respectively (see methods and Supplementary Table 7). None was present in 2 o 3 samples.

Annotation of genomic features showed that 8 of the 12 validated SVs detected by NanomonSV overlap with genes and that four of these directly affect coding sequences (a large deletion including IL15, SETD7 and MAML3; three medium-size deletion that affect respectively, TUSC7, ADCY7, CAV2, Supplementary Table 7).

Analyses of putative involvement in cancer revealed a potential role only for the tumor-suppressors TUSC7[37] and SETD7[38]. TUSC7 was associated to chemoresistance in one single study in

esophageal squamous cell carcinomas[39]. Allelic-fraction analyses of the TUSC7 deletion, however, showed relatively-low VAF (36%, Supplementary Table 7), arguing against a role in the acquisition of a chemoresistance phenotype in this case. The other SVs showed variable VAFs (all <0.3).

In general, all SVs exerted no effecst on gene expression, with the exception of the large deletion of AML2 that include six genes all significantly under-expressed (ZNF330, MGST2, MAML3, SCOC, NAA15 and INPP4B, Supplementary Table 8). Analyses of gene mutations in the same three AML-pairs by WES gave similar results: low numbers of relapse-specific mutations (11, 6 and 2 in the UD5, UD10 and AML2 samples), all mutations present at sub-clonal frequencies (from 10 to 40%), none functionally related to the drug-resistance phenotype (Supplementary Table 9).

Thus, relapse AML-samples are characterized by a few patient-specific SVs and DNA mutations that are unlikely to be involved in the acquisition of the drug-resistant phenotype.

## Discussion

We report the first WGS concomitant analyses of genetic and epigenetic alterations of human cancer using high-coverage nanopore data. To detect epigenomic modifications, we developed a novel computational method (PoreMeth), which allowed assignment of methylation to >99% CpGs of the human genome, with an average density of 1 CpG/100 bp. Most notably, ~40% of DMRs felt within low-density CpG regions (2 CpG/100 bp). As a reference, NGS-based RRBS or WGBS approaches generally identify DMRs in regions with CpG density ≥3 or ≥2 CpGs/100 bp, corresponding to ~20 or ~50% of the genome respectively[6]. Thus, PoreMeth analyses of whole-genome long-read sequences provided an unprecedented resolution of the cancer methylome. Analyses of differential expression identified several thousands of DEGs in the relapsed AMLs, largely overlapping across the three patients. As previously reported[28], methylation nearby TSSs correlated negatively with gene expression, much less for 3'UTR and enhancers and not at all for gene-bodies. Notably, however, the vast majority of differentially-methylated genes were not differentially expressed, as previously reported[40], suggesting that a large fraction of the aberrant DNA-methylation observed in our samples is a merely-passenger event that accompanies AML evolution, with weak or no effect on the relapsed/chemoresistance phenotype. Consistently, gene/pathway over-representation analyses of differentially-methylated and expressed genes showed no enrichment of any relevant gene set, with the exception of those carrying hyper-methylated DMRs overlapping sparse CpGs at gene-bodies, which showed significant over-representation of TFs. Surprisingly, analyses of putative targets of these TFs revealed the existence of large gene-regulatory networks (GRNs) made of over- and under-expressed genes, largely overlapping among the three samples and highly-enriched for chemoresistance genes. The GRNs were supported by few and largely patient-specific TFs: NCOR2, CUX1, ARID3A, ETV6, RERE, TFDP1 in the UD5 sample, RARa, BRF1 in UD10, and CUX1, NCOR2, RREB1, GTF2I, ZBTB16 in AML2. These are critical regulators of cell fate and differentiation in hematopoietic cells (e.g., ARID3A, RERE), are frequently mutated in AMLs (e.g., CUX1, ETV6, RARa), and posses a dual function of transcription activator or repressor (CUX1, ARID3A, ETV6, RARa, RREB1, GTF2I)[41–45] which is consistent with the presence in the GRNs of both over- and under-expressed genes. Strikingly, alone, these twelve TFs support 2263, 1798 and 947 GRN-DEGs, which represent 55, 38 and 50% of all DEGs, for UD5 UD10 and AML2 respectively.

Analyses of GRN-associated chemo-resistance genes revealed multiple gene de-regulations consistent with activation of different mechanisms of chemotherapy-resistance in the relapsed AMLs, including increased drug efflux, reduced drug uptake, increase pH in the extracellular environment and intracellular drug inactivation. TF gene-targets also showed over-representation of several pro-survival intracellular signaling pathways that have been implicated in both tumor development and chemo-therapy resistance (decreased apoptosis; enhanced DNA repair; activation of the TGF-b, p38 MAPK stress-response and phosphoinositide 3-kinase (PI3K) pathways; activation of Wnt/-catenin signaling cascade and the crosstalk STAT signaling pathway) (see Supplementary Data 3). Thus, hyper-methylation of sparse CpGs at gene bodies of few and patient-specific master TFs is the main source of gene deregulation in the chemo-resistant phenotype.

Distinct mechanisms of aberrant methylation may be operative at CpG islands or sparse CpGs. The occurrence of hyper-methylated DMRs overlapping CpG islands was significantly over-represented in all genic and regulatory elements, at variance with hyper-DMRs at sparse CpGs, which were instead characterized by high levels of randomness and heterogeneity (methylation entropy). Moreover, overlaps across the three samples was larger for genes with DMRs at CpG islands (20-60%), as compared to genes with DMRs at sparse CpGs (5-15%). Thus, formation of hyper-methylated DMRs at CpG islands or sparse CpGs is the consequence of highly ordered or stochastic processes, respectively, suggesting the involvement of active, enzymatically-driven vs random processes of DNA methylation.

Patterns of CpG methylation are established by two partially-redundant de novo DNA methyltransferases, DNMT3A and DNMT3B[46]. It has been recently reported that CpG-island hypermethylation in AMLs is mediated by DNMT3A and that also occurs in normal hematopoietic progenitors in response to cytokine-induced hyper-proliferation, implying that it is not required for myeloid leukemogenesis and merely reflects activation of a cell-cycle checkpoint earlier during leukemogenesis[47]. Most genes with CpG-island hypermethylation were not expressed in AMLs[47], as we observed in our relapsed AMLs, suggesting that they represent passenger epigenetic mutations for the chemoresistance phenotype.

Genome-scale mapping of CpG methylation-kinetics revealed highly-variable and context-specific activities of DNA-methylation enzymes, with high methylation turnover in transcribed gene-bodies and reduced de novo and maintenance methylation in CpGs at active regulatory regions[48]. Maintenance of CpG methylation during mitosis is mediated by the DNMT1 methyltransferase, which targets hemi-methylated DNA and catalyzes methylation during and after DNA replication[49]. The maintenance activity of DNMT1, however, is imprecise, thus leading to accumulation of spontaneous 'epimutations'[50]. DNMT1, however, is catalytically active also on unmethylated DNA and has been recently shown to possess de novo methylation activity[50–52]. Notably, DNMT1 and UHRF1, a E3 ligase that mediates DNA targeting of DNMT1, are widely overexpressed in cancer[53], including AMLs[54], suggesting that aberrant expression of DNMT1 may also generate CpG hyper-methylation in cancer cells, as shown in model systems[55]. We propose a model whereby accumulation in the relapsing AMLs of hyper- or hypo-methylation at sparse CpGs of gene-bodies is the consequence of the selective pressure of chemotherapy on the epigenomic heterogeneity of primary leukemias, as generated by maintenance failures and aberrant expression of DNMT1 (epigenetic-instability at CpG-poor regions). Consistently, DNMT1 is was significantly under-expressed in all three patients, contrary while

DNMT3A and DNMT3B did not show expression changes across the three samples (Supplementary Table 10).

In principle, epigenetic alterations are reversible, thus opening the possibility of pharmacological modulation of epigenetic heterogeneity to prevent the emergence of drug resistance in cancer cells.

## Materials and methods

**Samples preparation**. Diagnosis and relapse samples were collected for each AML patient in two different Italian Institutions: Azienda Socio-Sanitaria Territoriale Papa Giovanni XXIII, Bergamo, and Azienda Sanitaria Universitaria Integrata di Udine, Udine, as part of research projects approved by the relevant Institutional Ethical Committees. Mononuclear cells (MNCs) were isolated by density gradient centrifugation using Ficoll-Paque Plus from bone marrow samples of AML patients with blasts infiltration around 70%. After centrifugation and recovery of the MNCs, the remaining erythrocytes were lysed using lysis buffer (155 mM NH4Cl, 10 mM KHCO3, 0.1 mM Na2EDTA). DNA was extracted from the isolated MNCs. The existence of SNVs with VAF > 0.4 in both T and R samples (Supplementary Table 8) allows to estimate that tumor purity can be considered >80%.

**Long read sequencing, filtering and alignment**. DNA from each of three pairs of matched AMLs was sequenced on the GridION X5 instrument with a run-time of 48 h by using five or six individual R9.4 flow cells for each sample (see Supplemental Material). The reads generated by each run were base-called using Guppy v2.0.2 and quality assessment was performed with NanoR[56] and PyPore[57]. High-quality pass reads were then aligned against the human reference genome (hg19) with minimap2[58].

**Methylation**. Occurrence of 5mC was determined with Nanopolish (v. 0.8.5) on the raw signals from Nanopore's sequencers, by first tracking signal-data for each base called read in FASTQ, ("nanopolish index"), then aligning files in BAM format and calling methylation at reference CpGs with the "nanopolish call-methylation". Any given site was considered methylated or unmethylated when the log likelihood ratio was > or < 2.5, respectively. Methylation frequencies ($\beta$) at reference CpGs were then summarized using the "calculate methylation frequency" script, which clusters nearby CpGs to produce the final calls (since nanopore signals depend on multiple bases).

**PoreMeth**. SLM are a special class of hidden markov models in which sequential observations $x = (x_1, \ldots, x_i, \ldots, x_N)$ are considered to be realizations of the sum of two independent stochastic processes $x_i = m_i + \epsilon_i$, where $m_i$ is the unobserved mean level and $\epsilon_i$ is normally distributed white noise.

The mean level $m_i$ does not change for long intervals and its duration follows a geometric distribution: the probability that $m_i$ takes a new value at any point $i$ is regulated by the parameter $\eta$ and when it changes, $m_i$ is incremented by the normal random variable $\delta_i$ ($\delta_i \sim N(0, \sigma_\mu^2)$). To take into account the sparse distribution of CpGs along the genome, as in ref. [59] we extended the classical SLM to an heterogeneous form[59], where $\eta$ depends on this distance between consecutive CpGs ($d_i$) with the following formula:

$$\eta(d_i) = \frac{1}{2} \cdot \theta + \left( \left( \frac{1}{2} - \theta \right) \cdot exp \left[ \frac{log(\theta)}{\frac{d_i}{d_{Norm}}} \right] \right) \tag{1}$$

Moreover, since $\Delta\beta$ takes values in the range $[-1,1]$, we modeled $\epsilon_i$ with truncated gaussian distributions with upper and lower bound 1 and $-1$ respectively ($\epsilon_i \sim N^1_{-1}(0, \sigma_\epsilon^2)$).

To estimate the parameters of the heterogeneous truncated-gaussian shifting level model (HTGSLM), and consequently segment the $\Delta\beta$ profiles, we developed a two-step algorithm in which we first calculate mean and variances and we then apply the Viterbi algorithm.

After segmentation, each segment is tested for differential methylation by using the Wilcoxon rank-sum test on the $\beta$ values of test and control samples. PoreMeth is freely available at https://sourceforge.net/projects/poremethtool/.

**Annotation**. DMRs were annotated to Refseq gene models, CpG islands, DHSs, TFBS and enhancers (predicted by FANTOM5) by using annotatr version 1.20[60]. Refseq gene models annotations include from 1 to 5 Kb (1to5Kb) upstream of the TSS, the promoter (1 kb upstream of the TSS), the 5' untranslated region (5'UTR), first exons, exons, introns, 3'UTR, and intergenic regions. FANTOM5 permissive enhancers were determined from bi-directional CAGE transcription as in ref. [61]. In all the analyses we performed, 5' regulatory regions (Reg) include 1to5Kb, promoter, 5'UTR and first exon while gene bodies (GB) include internal exons and introns. Deletions and duplications were annotated by including all the genomic elements within the interval between the two breakends.

**Over-representation analysis (ORA)**. Pathways for ORA were selected by the network of 'PATHWAYS IN CANCER' of KEGG database and the Oncogenic Signaling Pathways in The Cancer Genome Atlas[62]. Gene lists of these pathways were downloaded from https://www.kegg.jp/kegg/download/(KEGG), COSMIC genes from https://cancer.sanger.ac.uk/cosmic/file_download, GEAR genes from http://gear.comp-sysbio.org, RegNetwork from http://regnetworkweb.org/. ORA was performed by using fisher-exact test[63] using the list of all UCSC genes as background.

**RNA-seq**. RNA quality was evaluated using Agilent RNA 6000 Nano Kit on the Agilent 2100 Bioanalyzer (Agilent Technologies, Santa Clara, USA). RNAseq libraries were prepared using TruSeq Stranded Total RNA Library Prep Gold (Illumina, San Diego, CA, USA) according to the manufacturer's protocol and sequenced using 50bp paired end sequencing mode on Illumina Novaseq 6000 platform (Illumina, San Diego, CA, USA). To perform counts of transcripts from paired-end reads we used Salmon[64] v. 0.14.1, in mapping-based mode and the reference transcriptome GRCh37 from Ensembl. The resulting qf files were imported in R by the package tximport and the transcripts were collapsed to the genes by EnsDb.Hsapiens.v75 (v. 2.99.0) package to perform counts normalization and the statistical analysis with DESeq2 (v. 1.30.1). Results with an adjusted p-value, scored by Benjamin-Hopkins formula, greater than 0.05 were filtered. Direct TF-DNA interactions between DM-DE-TFs and DE-TGs were downloaded from UniBind database (unibind.uio.no), considering only robust interactions. RNA-seq data for the validation cohorts were downloaded from ArrayExpress for[35] and from SRA for[34]. Data were analyzed by applying the same RNA-seq pipeline described above for our dataset.

**Observed vs expected analysis**. The observed vs expected ratio for the occurrence of a DMR in any given genomic region was calculated using the following formula:

$$\frac{O_{GF}^{DMRs}}{E_{GF}^{DMRs}} = log_2 \left( \frac{F_{GF}^{Obs}}{F_{GF}^{Exp}} \right) = log_2 \left( \frac{\frac{Size_{GF}^{DMR}}{Size_{Total}^{DMR}}}{\frac{Size_{GF}^{Genome}}{Size_{Total}^{Genome}}} \right) \tag{2}$$

where $Size_{GF}^{DMR}$ is the total size of DMRs overlapping a genomic feature $GF$, $Size_{Total}^{DMR}$ is the total size of identified DMRs, $Size_{GF}^{Genome}$ is the total size of a genomic feature $GF$ in the genome and $Size_{Total}^{Genome}$ is the total size of the genome.

Statistical significance of observed vs expected ratio for the occurrence of DMRs in any given genomic element was calculated by using montecarlo simulations. In brief, for each sample, we randomly generated $n$ genomic segments (where $n$ is the number of hypo- or hyper-methylated DMRs found by Poremeth) with the same size distribution of real datasets and we then calculated $\frac{O_{GF}^{DMRs}}{E_{GF}^{DMRs}}$. The P-value is the relative ranking of the real $\frac{O_{GF}^{DMRs}}{E_{GF}^{DMRs}}$ among the sample values from the Monte Carlo simulation. For each analysis we performed 1 million simulations.

**Methylation entropy**. Methylation entropy (ME) was calculated by using the formula introduced in ref. [36]:

$$ME = \frac{e}{b} \Sigma \left( -\frac{n_i}{N} + Log \frac{n_i}{N} \right) \tag{3}$$

where $e$ is entropy for code bit, $b$ is the number of CpG sites, $n_i$ is the occurrence of methylation pattern $i$ and $N$ is the total number of reads overlapping the $b$ CpG sites. For each genomic feature, ME was calculated by using $b = 3$ and averaged across all the CpG sites within the feature.

**Structural variant detection and validation**. Relapse-specific SVs detection was performed by using and NanomonSV. We downloaded NanomonSV version 0.4.0 from https://github.com/friend1ws/nanomonsv and we applied it to the three AML pairs with default parameter settings by using relapsed samples as tests and tumor samples as controls. SVs with VAF (ratio between reads supporting SV and total number of reads) <0.1 were filtered out. SVs detected by NanomonSV were validated by visual inspection by using IGV version 2.9.4 and Samplot version (see Supplemental Material for more details).

**AML Illumina WES analysis**. Genomic DNA was extracted from cryopreserved bone marrow (BM) mononuclear cells using QIAGEN's AllPrep DNA/RNA kit. Libraries were prepared following SureSelectXT Low Input Reagent Kits (Agilent Technologies) procedure for dual-indexing, target enrichment and capture. Paired-end sequencing was performed on a Novaseq instrument at ~200x coverage. Somatic variants were called against germline reference (remission BM, UD5 and UD10; BM derived fibroblasts, AML2) using state-of-the-art computational pipelines, such as Mutect, Pindel and Somatic_Indel_detector. Only mutations with variant allelic frequency ≥10% were considered in the presented analysis.

**Reporting summary**. Further information on research design is available in the Nature Portfolio Reporting Summary linked to this article.

## Data availability

Nanopore sequencing data in fastq format have been deposited in the NCBI Sequence Read Archive under accession number PRJNA879930. RNASeq in fastq format have been deposited in the NCBI Sequence Read Archive under accession number PRJNA879971. RNASeq normalized counts and Methylation frequency data are available at Gene Expression Omnibus under accession numbers GSE213686.

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

## Acknowledgements

This work was supported by the Associazione Italiana per la Ricerca sul Cancro to Alb. M. and P.G.P. (AIRC Investigator Grants 20307 and 20162, respectively).

## Author contributions

Alb.M. and P.G.P.designed the study and wrote the manuscript. A.R., A.C., C.R. and E.C. contributed to the patients enrollment and were responsible for the appropriate follow-up. Ale.M., C.R. and G.F. performed the DNA and RNA sequencing experiments. Alb.M. developed PoreMeth. Alb.M., G.M. R.S. and D.B. performed all bioinformatic analyses. Alb.M., P.G.P., G.M., L.M. and C.C. contributed to data interpretation. All the authors reviewed and edited the manuscript.

## Competing interests

All the authors declare no competing interests.
