## [Peer Review File · Communications Biology]

Reviewers' comments:

Reviewer #1 (Remarks to the Author):

In this manuscript, the authors conducted whole genome sequencing of samples from three AMLs patients at diagnosis and relapse using nanopore sequencer. They performed methylation calling from nanopore data using nanopolish. They developed PoreMeth a tool for detection of differentially methylated regions (DMRs) and applied the tool to detect DMRs between each pair at diagnosis and relapse. They found that ~40% of the DMRs are located on sparse CpGs and most DMRs were hyper-methylated. They also performed RNA-seq of the same samples and collected differentially expressed genes (DEGs) between each pair. They indicated that only 15% of differentially methylated genes (DMGs) show differential expression. 40-70% of DEGs were shared between patients. Transcriptional factors and cancer-related genes were over-represented in DEGs and DMGs. They also showed that TFs are over-represented in hyper-methylated DMGs at sparse CpGs. They suggested that a few TFs, which are regulated by methylation at sparse CpG, might be regulators of gene regulatory network in relapse. They also performed detection of SVs and point mutation using nanopore data and exome data, respectively. Although there are some unclear points, I think this study is potentially interesting and the data would be valuable. There are, however, several points to be considered for the possibility of further improvements:

1. I could not find the description about preparation of tumor cells. DNA methylation analysis is greatly affected by contamination of other normal cells. It is important to explain in detail how the cells are prepared. If the samples contains plenty of normal cells, the data may be difficult to analyze.
2. In order to assess methylation status, it is better to compare the average of methylation rates of all CpG sites and to perform a correlation analysis of methylation rates per CpG site between samples at diagnosis and relapse at first.
3. Expression levels of components of machineries for DNA methylation and demethylation in each sample should be investigated using RNA-seq data.
4. Throughout the manuscript, we felt that not enough information on the expression and methylation patterns of individual genes was presented. It would be easier for readers to evaluate the results if expression and methylation patterns of some genes, especially DM-DE-TFs, were visualized using IGVs for diagnosis and relapse.
5. The authors showed 12 of relapse-specific SVs and its VAFs. The significance of these SVs is still unclear. The SVs may affect the expression levels of the genes around them. Therefore, it is better to validate this influence using RNA-seq data.

Reviewer #2 (Remarks to the Author):

In this manuscript, Magi et al. analyzed DNA methylation patterns of paired AML samples at diagnosis and relapse using a nanopore sequencer and their developed method PoreMeth. They identified DMRs in relapsed samples and described their features in each of the CpG-resolutions, such as CGIs and sparse CpGs, and genomic functional domains, such as regulatory regions/elements and gene bodies. They also identified a few transcription factors which were under-expressed and hyper-methylated in sparse CpGs. The TFs would control large regulatory networks which could be associated with chemoresistance. The reviewer thinks that the developed method may be useful for detection of long DMRs consisting of sparse CpGs. However, biological relevance of the identified events would not be clearly elucidated and validated. There are points that need to be addressed as below;

Points;

1. The authors identified TFs with hyper-methylation of sparse CpGs and their regulating networks. They should conduct validation analysis. For example, they need to investigate whether differentially methylated CpGs really affect the TF expression, and whether the identified TFs bind and regulate the downstream network in relapsed AML cells. Moreover, the authors need to verify existence of the networks in other AML samples as a validation cohort.
2. The data from R9.4 flow cells still have approximately 10% sequencing errors. The authors need to show and discuss possible errors in methylation analysis of sparse CpGs. IGV visualization of reads would be helpful for quality control.
3. The authors should describe materials and patient information in the Materials and Methods section. They should add the information on ethical approval for this study.
4. p.5: ERRBS -> RRBS?

Replies to Reviewers' comments:

Reviewer #1 (Remarks to the Author):

In this manuscript, the authors conducted whole genome sequencing of samples from three AMLs patients at diagnosis and relapse using nanopore sequencer. They performed methylation calling from nanopore data using nanopolish. They developed PoreMeth a tool for detection of differentially methylated regions (DMRs) and applied the tool to detect DMRs between each pair at diagnosis and relapse. They found that ~40% of the DMRs are located on sparse CpGs and most DMRs were hyper-methylated. They also performed RNA-seq of the same samples and collected differentially expressed genes (DEGs) between each pair. They indicated that only 15% of differentially methylated genes (DMGs) show differential expression. 40-70% of DEGs were shared between patients. Transcriptional factors and cancer-related genes were over-represented in DEGs and DMGs. They also showed that TFs are over-represented in hyper-methylated DMGs at sparse CpGs. They suggested that a few TFs, which are regulated by methylation at sparse CpG, might be regulators of gene regulatory network in relapse. They also performed detection of SVs and point mutation using nanopore data and exome data, respectively. Although there are some unclear points, I think this study is potentially interesting and the data would be valuable. There are, however, several points to be considered for the possibility of further improvements:

Specific points:

1. I could not find the description about preparation of tumor cells. DNA methylation analysis is greatly affected by contamination of other normal cells. It is important to explain in detail how the cells are prepared. If the samples contains plenty of normal cells, the data may be difficult to analyze.

We added a paragraph on tumor cells preparation in Materials and Methods section.

2. In order to assess methylation status, it is better to compare the average of methylation rates of all CpG sites and to perform a correlation analysis of methylation rates per CpG site between samples at diagnosis and relapse at first.

As suggested, we calculated the average of methylation rates of all CpG sites for each sample, and performed a correlation analysis of methylation rates per CpG site between samples at diagnosis and relapse. The results of these analyses are reported in supplemental figure 3 and 4 and briefly discussed in results section of the manuscript.

3. Expression levels of components of machineries for DNA methylation and demethylation in each sample should be investigated using RNA-seq data.

As requested, we analysed expression levels of all the components of the methylation machinery (DNMT1, DNMT3A, DNMT3B, TET1, TET2 and TET3), now reported in Table 10 of Supplemental Material. We commented these results in the Discussion Section to support our "epigenetic-instability at CpG-poor regions" model.

4. Throughout the manuscript, we felt that not enough information on the expression and methylation patterns of individual genes was presented. It would be easier for readers to evaluate the results if expression and methylation patterns of some genes, especially DM-DE-TFs, were visualized using IGVs for diagnosis and relapse.

We agree. Not- enough information was presented on the expression and methylation patterns of individual genes. Thus, we added IGVs plots of methylation patterns and expression of DM-DE-TFs with DMRs in low-density CpG regions at gene-bodies. (Figure 1-13 of Supplemental File 5).

5. The authors showed 12 of relapse-specific SVs and its VAFs. The significance of these SVs is still unclear. The SVs may affect the expression levels of the genes around them. Therefore, it is better to validate this influence using RNA-seq data.

As requested, we analysed expression levels for all the genes affected by SVs, by DESeq2. Results are shown in Table 9 of Supplemental Material.

Reviewer #2 (Remarks to the Author):

In this manuscript, Magi et al. analyzed DNA methylation patterns of paired AML samples at diagnosis and relapse using a nanopore sequencer and their developed method PoreMeth. They identified DMRs in relapsed samples and described their features in each of the CpG-resolutions, such as CGIs and sparse CpGs, and genomic functional domains, such as regulatory regions/elements and gene bodies. They also identified a few transcription factors which were under-expressed and hyper-methylated in sparse CpGs. The TFs would control large regulatory networks which could be associated with chemoresistance. The reviewer thinks that the developed method may be useful for detection of long DMRs consisting of sparse CpGs. However, biological relevance of the identified events would not be clearly elucidated and validated. There are points that need to be addressed as below;

Specific Points:

1. The authors identified TFs with hyper-methylation of sparse CpGs and their regulating networks. They should conduct validation analysis. For example, they need to investigate whether differentially methylated CpGs really affect the TF expression, and whether the identified TFs bind and regulate the downstream network in relapsed AML cells. Moreover, the authors need to verify existence of the networks in other AML samples as a validation cohort.

We interrogated the UniBind database, a comprehensive map of direct transcription factor (TF) – DNA interactions in the genome for nine different species. We only considered "robust" interactions for the human genome. Of the twelve DM-DE-TFs with DMRs in sparse CpG regions at gene-bodies, five are present in UniBind. Strikingly, all five DM-DE-TFs showed direct TF–DNA interactions in more than 50% of the DE-TGs of our first-level regulatory networks, strongly supporting the role of these TFs in deregulating the inferred GRNs. We added a Table on Supplemental Material with these informations (Table 6 of Supplemental Material). Finally, to verify the existence of the same networks in other AML samples, we compared DEGs of our analyses with those identified by other two papers and found statistically- significant overlap between first- and second-level GRNs and all DEGs with DEGs previously identified in the two independent cohorts, thus demonstrating the existence of the same deregulated modules in other AML patients. We added these results in Supplemental Figure 42.

2. The data from R9.4 flow cells still have approximately 10% sequencing errors. The authors need to show and discuss possible errors in methylation analysis of sparse CpGs. IGV visualization of reads would be helpful for quality control.

We agree with the reviewer that the high sequencing error rate of R9.4 flow cells might affect estimation of methylation frequency at single CpGs (Beta Values). However, our approach consists in searching consecutive CpGs with increased or decreased delta-beta values. Thus, identification of DMRs based on many consecutive CpGs (more than 5) strongly mitigates the

effects of sequencing errors on the estimation of methylation frequency and of methylation-frequency differences.

Concerning the impact of the error rate on sparse CpGs, we did not observe any specific differences with high-density CpG regions. Following reviewer's suggestions, we plotted all DMRs affecting DM-DE-TFs on gene-bodies in Figure 1-13 of Supplemental File 6.

3. The authors should describe materials and patient information in the Materials and Methods section. They should add the information on ethical approval for this study.

We added a new section (Samples preparation) in Material and Methods with patient information.

4. p.5: ERRBS -> RRBS?

We corrected ERRBS with RRBS.

REVIEWERS' COMMENTS:

Reviewer #1 (Remarks to the Author):

The authors have addressed all of my comments. The manuscript improved well.

Reviewer #2 (Remarks to the Author):

In the revision, the authors have improved the manuscript and addressed the previous concerns including validation analysis of interaction and QC for methylation call. The reviewer just recommends that unmethylated CpGs are also plotted in a color (for example, in blue) in IGV plots of Supplemental file 5 because we could not distinguish unmethylated CpG from undetermined one in grey color.

In the revision, the authors have improved the manuscript and addressed the previous concerns including validation analysis of interaction and QC for methylation call.

The reviewer just recommends that unmethylated CpGs are also plotted in a color (for example, in blue) in IGV plots of Supplemental file 5 because we could not distinguish unmethylated CpG from undetermined one in grey color.

As requested by the Reviewer we plotted unmethylated CpGs in blue.